# A Novel Approach to Pod Count Estimation Using a Depth Camera in Support of Soybean Breeding Applications

**DOI:** 10.3390/s23146506

**Published:** 2023-07-18

**Authors:** Jithin Mathew, Nadia Delavarpour, Carrie Miranda, John Stenger, Zhao Zhang, Justice Aduteye, Paulo Flores

**Affiliations:** 1Agricultural and Biosystems Engineering Department, North Dakota State University, Fargo, ND 58105, USA; jithin.mathew@ndsu.edu (J.M.); paulo.flores@ndsu.edu (P.F.); 2Department of Plant Sciences, North Dakota State University, Fargo, ND 58105, USA; carrie.miranda@ndsu.edu; 3North Dakota Agricultural Weather Network, School of Natural Resource Sciences, North Dakota State University, Fargo, ND 58105, USA; john.stenger@ndsu.edu; 4College of Information and Electrical Engineering, China Agricultural University, Beijing 100083, China; zhaozhangcau@cau.edu.cn; 5Department of Agronomy, Earth University, San Jose 4442-1000, Costa Rica; justice.aduteye@ndsu.edu

**Keywords:** background segmentation, computer vision, deep learning, depth camera, high throughput phenotyping, machine vision, soybean pod-counting

## Abstract

Improving soybean (*Glycine max* L. (Merr.)) yield is crucial for strengthening national food security. Predicting soybean yield is essential to maximize the potential of crop varieties. Non-destructive methods are needed to estimate yield before crop maturity. Various approaches, including the pod-count method, have been used to predict soybean yield, but they often face issues with the crop background color. To address this challenge, we explored the application of a depth camera to real-time filtering of RGB images, aiming to enhance the performance of the pod-counting classification model. Additionally, this study aimed to compare object detection models (YOLOV7 and YOLOv7-E6E) and select the most suitable deep learning (DL) model for counting soybean pods. After identifying the best architecture, we conducted a comparative analysis of the model’s performance by training the DL model with and without background removal from images. Results demonstrated that removing the background using a depth camera improved YOLOv7’s pod detection performance by 10.2% precision, 16.4% recall, 13.8% mAP@50, and 17.7% mAP@0.5:0.95 score compared to when the background was present. Using a depth camera and the YOLOv7 algorithm for pod detection and counting yielded a mAP@0.5 of 93.4% and mAP@0.5:0.95 of 83.9%. These results indicated a significant improvement in the DL model’s performance when the background was segmented, and a reasonably larger dataset was used to train YOLOv7.

## 1. Introduction

Soybean (*Glycine max* L. (Merr.)) is the world’s most widely grown leguminous crop [1] and represents one of the world’s most important sources of protein and edible oil [2,3]. Soybean has numerous economic advantages because of its high nutritional value, raw material for edible oil processing factories, and soil-enriching potential due to symbiotic N2 fixation [4]. Moreover, because of the high concentration of protein content, soybean is an excellent nutritional alternative to animal-source proteins [5]. Due to the importance of this crop, understanding the underlying genetic mechanism behind the agronomic traits is very crucial to improving its yield potential.

One way to improve soybean yields is through the development of higher-yield varieties using plant breeding techniques. High-throughput field phenotyping is one of these techniques that provides an affordable and efficient tool to evaluate important agronomic traits in a large number of soybean genotypes. During the plant breeding process, agronomic and seed quality attributes can be improved by crossing phenotypically superior crop cultivars and selecting improved offspring in each cycle [6]. Desirable traits in soybean breeding lines include high yield, resistance to diseases, and abiotic stresses [7,8,9]. Among the desirable traits, crop yield is considered a primary trait for selecting superior genotypes in crop breeding programs from numerous breeding lines in multiple years and locations [10]. Crop yield is affected by variables other than genetics, including weather conditions, soil type, seed variety, fertilizers, etc. In more general terms, the yield production depends on the crop genotype and its interaction with the environment [11,12,13].

In conventional methods of soybean breeding based on manual sampling techniques, a large number of progenies are made within a given population derived from a designed cross, advanced for several generation, and then evaluated for agronomic characteristics in single- or double-row plots. Based on visual yield estimation, a certain percentage of progenies (usually 10%) is selected to grow at multiple locations to further select a smaller group of genotypes that perform better under a variety of environmental conditions. The goal of breeders can be outlined through an equation used to track breeding progress, which is called the breeder’s Equation (Equation 1):(1)$$\frac{\Delta \mu }{t}=\frac{\left(h^{2}\right)\left(i\right)\left(\sigma_{p}\right)}{t},$$
where Δμ is the change in the mean value of the population between breeding cycles, *t* is the time per cycle, $h^{2}$ is the narrow sense heritability estimate, *i* is the standardized selection differential, and $\sigma_{p}$ is the standard deviation among all phenotypes [14]. One key component relating to genetic gain that breeders control through field evaluations is the accuracy of selection and its impacts on the realized narrow sense heritability of a trait. Narrow sense heritability is the ratio of additive genetic variance (heritable variance) to phenotypic variance [15] given in Equation (Equation 2):(2)$$\;h^{2}=\frac{\sigma_{a}^{2}}{\sigma_{p}^{2}}.$$

Phenotypic variance can be caluculated using Equation (Equation 3), which is the sum of additive genetic variance (a), non-additive genetic variance (nag), genotype-by-environment interaction variance (ge) and non-genotype-related variances such as environmental (e) and residual error (error).
(3)$$\;\sigma_{p}^{2}=\sigma_{a}^{2}+\sigma_{nag}^{2}+\sigma_{ge}^{2}+\sigma_{e}^{2}+\sigma_{error}^{2}.$$

A rapid phenotyping method can impact breeding progress. Reduction in the time required to conduct evaluations can result in the ability to increase population size allowing opportunities to enact greater selection intensity leading to positive impacts on genetic gain. Overall, reductions in the non-genetically based errors that contribute to increases in a breeder’s estimation of phenotypic variance result in a net increase in genetic gain as the heritable value of an individual is more accurately accounted for as outside noise is reduced. Additionally, expediting data collection could facilitate the realization of additional gains as increased population sizes are feasibly evaluated with accuracy.

The conventional soybean phenotyping procedure is labor-intensive, inefficient, and slow [16]. Moreover, the accuracy and capacity of standard procedures for manual soybean phenotyping could be prone to subjectivity due to yield visual assessment [17]. Thus, objective prediction methods are needed to predict soybean yield in progenies or early breeding cycles over large areas at low cost [18,19]. Identifying high-performing soybean varieties (efficiently and accurately) early in the breeding pipeline could reduce costs and increase genetic gains for new lines [20]. Moreover, methods that reduce the non-genetic error variance observed in field observations and contribute to increasing selection accuracy through the realization of increased heritability (thus improving genetic gain) are desired.

Digital high-throughput phenotyping methods can consistently and rapidly acquire extensive data regarding crop yield. Various approaches have been applied for crop yield prediction, including crop-growth modeling coupled with environmental factors (i.e., hydrological, meteorological, and soil factors) [21], remote-sensing-based methods (i.e., satellite and Unmanned Aerial Systems (UASs)) [19,22], and proximal sensing (i.e., robots, smartphone cameras) [23]. Spectral, structural, thermal, and textural information (or features) extracted from remote and proximal sensing tools, including RGB, multispectral, hyperspectral, and thermal sensors, have been used for estimation of different crops’ yield [24,25,26,27,28,29,30,31].

Aerial imagery crop yield estimation methods are commonly based on the high correlation between the crop yield and features representing important crop traits including plant height, canopy cover, chlorophyll content, stress symptoms, maturity dates, the vegetation index (VI) taken at a specific phenological stage, canopy thermal information, and canopy structure (i.e., canopy height) [32,33,34,35,36,37,38,39,40]. The effect direction of these features could be different, e.g., soybean yield is positively correlated with moderate resolution imaging spectroradiometer (MODIS) NDVI and negatively correlated with daytime land surface temperature (LST) [41].

Recently, Machine Learning (ML) has provided powerful tools to plant breeders to conduct in-season seed yield prediction [42]. These models can automatically learn from data using a multi-layer architecture, learn from the hierarchical outputs of the previous layers, and deal with non-linearities between crop traits and yield estimation [31,43,44]. ML algorithms such as random forest and neural networks have been widely used to predict crop yield using data collected remotely [5,11,45,46,47,48,49,50]. It should be noted that the model accuracy is affected by the dates of the predictions [44], so it is important to consider this factor while making soybean yield predictions using satellite imagery. A polynomial regression model or a combination of different algorithms could overcome the limitations associated with the conventional linear regression model in soybean yield prediction [51]. These advances in ML methods and the earnest effort to collect large datasets is commendable and has a positive role in numerous scenarios; however, these approaches do not work for plant breeding programs of all sizes, geographical regions, and crops [52,53].

With application programming interfaces (APIs), e.g., TensorFlow and Keras, deep learning (DL) models, such as deep neural networks (DNNs), convolutional neural networks (CNNs), and recurrent neural networks (RNNs), it is possible to estimate crop yields with higher accuracy [54]. Providing a large amount of data, DL accuracies can be further improved compared with various regression and classification tasks [55,56,57,58]. It is important to determine the algorithm (e.g., Partial Least Squares Regression (PLSR), Random Forest Regression (RFR), Support Vector Regression (SVR), input-level feature-fusion-based DNN (DNN-F1), intermediate-level feature-fusion-based DNN (DNN-F2)) for the crop features that are considered in particular environmental conditions to predict soybean yield [59]. These techniques could provide deterministic estimates and do not account for the uncertainties involved in model predictions. A combination of of CNN models and integration of different algorithms have been used to overcome this problem [10,60]. Also, similar to satellite imagery, soybean stage of growth could significantly affect the accuracy of predictions and should be considered [10].

Although yield estimation based on aerial imagery has shown great potential in soybean yield estimation, these platforms only provide estimation for a broader region (county, state, country) and do not satisfy an in-filed real-time yield estimate for individual farmers and breeders. In addition, these methods are limited by low spatial resolution of collected images, atmospheric cloud conditions, and data acquisition frequency, especially for the application of plot-level assessment of breeding trials [61,62,63].

When field real-time yield estimation is required, pod count can be a useful indicator of yield potential [64]. In general, a higher number of pods per plant is associated with a higher yield. However, it is important to consider other factors, including the number of pods per plant, the number of seeds per pod, and the weight of each seed as well when predicting yield. Counting pods manually is an extremely laborious task, and the outcomes are susceptible to inaccuracies caused by worker fatigue [20,23,52,64]. Thus, proximal sensing tools and the possibility of collecting data from crops in fields are required to count the pods in a more efficient and accurate way. One method that is used to count soybean seeds is to collect a large dataset of soybean images scattered on a black background, mark the center of each seed in a soybean pod, and train the dataset to count soybean seeds automatically [23]. Since the datasets collected for this method do not contain any real-world field images of soybeans, the model is incapable of field real-time soybean seed counting. Another method that is developed to address the challenge of field real-time soybean seed counting includes collecting data from multiple sensors (fisheye camera, odometry, LiDAR, and GPS) by running a robot along an entire column at a time and training a DL model to detect and count pods in real time [20].

Since ML and DL models rely on detection of the target of interest on the images fed into the models, pod occlusion and background clutter of images (due to soybean plant architecture) could effectively reduce the accuracy of pod counting [52]. There have been several attempts to reduce the effects of occlusion and noisy images on the accuracy of yield prediction [65,66,67,68,69,70,71,72]. However, most of these methods only work for large plant organs (e.g., fruits) or when there are minimal occlusion issues. To deal with the problem of occlusion in soybeans, Riera et al. [52] developed a tracked robot with a mounted Logitech C920 camera to collect data from soybean crops from different angles. Also, for the control dataset, they used a trifold black background to remove background artifacts from the images. The results showed that in the control dataset with a trifold black background, the developed RetinaNet algorithm performed better compared to that for the in-field dataset. However, the usage of multi-view images as opposed to only-single view images only achieved a marginal improvement in yield estimation. The results of that study highlight the importance of using a method that removes the background of images to improve accuracy when performing automated soybean pod counting.

Although the study conducted by Riera et al. [52] showed a better result when the background was removed from the images, it is not feasible to isolate each and every plant in the field using a trifold black background for the purpose of imagery process. To overcome that issue, we investigated the capability of depth cameras in removing background beyond a threshold distance. The possibility of segmentation of region of interest using a stereo vision depth camera was investigated before, and the results showed the proposed method is able to segment desired target from the background successfully [73,74]. To the best of our knowledge, the approach of using a stereo vision depth camera to remove the background of soybean images has never been studied before. Thus, in this study, we analyzed the hypothesis of improving soybean pod counting by removing the image background using a depth camera coupled with a DL model (YOLOv7). For this purpose, we developed a platform to test the application of the depth camera along YOLOv7 algorithm in the soybean breeding test plots. The aim was to test whether removing noisy clutter background using a depth camera could result in a higher accuracy in distinguishing and counting pods compared with the case when the background of the images is not removed. The rest of the paper is organized as follows: Section 2 (Materials and Methods) presents the data collection and pre-processing steps along with the ML framework for pod counting and plant detection from plots. In Section 3 (Results), we present the performance of our proposed framework compared with the absence of background removal. In Section 4 (Discussion), we discuss the feasibility and promise of our approach to breeding programs. Finally, Section 5 summarizes the current paper and concludes the directions for future work.

## 2. Materials and Methods

### 2.1. Data Acquisition and Preprocessing

In this study for DL model development and validation, a field test dataset was collected from different soybean cultivars in a field with high environmental variability located in Casselton, ND, USA (Table 1). Images were captured using an Intel^@^ RealSense^TM^ Depth Camera D405 (©Intel Corporation, Santa Clara, CA, USA) for several rounds with varying light conditions during the summer of 2022. Depth accuracy of the camera at 50 cm is ±2%, its field of view is $87^{\circ }\times 58^{\circ }$, the covered depth distance is 7–50 cm, and maximum resolution is 1280 × 720.

The plot plant height ranged from 25 cm to 108 cm, with a median height of 70 cm and a standard deviation of 15.99 cm. A bicycle-like platform (Figure 1) was built to carry the camera between the soybean rows and to collect images from the side views of soybean lines in different plots (Figure 1). The camera was mounted on the platform at 44.4 cm from ground level to ensure that the full length of the soybean plants was imaged in each frame. The sensor placement on the platform was kept the same during the whole data collection, and all images were collected using the same camera angle and field of view. The platform was manually moved across the field during data collection, and a Python v3.9.11 script was used to automate the image capturing. To be able to predict an accurate number of pods per plot, it was ensured that images from the entire plot length were captured. Moreover, the images were collected at the full maturity (R8) stage of crop growth. A total of 35,082 images were collected from four different blocks of soybeans with different soybean cultivars. Figure 2 represents images of soybeans in the field before and after removing the background of an image with respect to the defined maximum covered depth (50 cm).

### 2.2. Image Preprocessing and Annotation

Prior to choosing images for annotation, images were filtered to discard the ones with no soybean pods to improve the overall quality of the dataset. To achieve this, we trained a VGG16 classifier to perform binomial classification, which retains the good images and removes the noisy ones. Also, an expert rater determined the start and end of the frame sequences for each plot in each pass. This was to ensure that all the images were of reasonable quality before building a dataset for ML. The number of video frames per plot ranged from 13 to 98, with a median of 38 frames per plot. Images from a total of 759 different plot rows (Table 1) were combined to generate a primary dataset containing 35,082 images. Following initial pre-processing, backgrounds were removed from all the images in the primary dataset. Two sub-dataset of 1000 images were generated by randomly picking images from the primary dataset, the first of which contained images with background removed and the other set contained the same images but with the background present. Background removed images were annotated using LabelImg software (https://github.com/tzutalin/labelImg, accessed on 21 January 2023) under the YOLO labeling format to meet YOLOv7 and YOLOv7-E6E model requirements. This resulted in three separate datasets, which are referred to as primary (the whole dataset with 35,082 segmented images), background (1000 random images from the primary dataset with background present), and no-background (the segmented images of the 1000 random images in the background dataset) datasets hereafter. The labels generated during the annotation of the non-background dataset were also used for the images in the background dataset. Finally, images of all three datasets were split into training and test sets in a 90:10 ratio.

### 2.3. YOLOv7 Object Detection Model

Since the first release of YOLOv1 [75], the YOLO family has undergone several major revisions and has shown a great potential in improving the visual object detection accuracy [76,77,78]. YOLOv1 was introduced as a Single-Shot Detector and effectively solved the slow reasoning speed of the Double-Shot Detector networks without sacrificing the detection accuracy. Since then, the lightweight and high accuracy of the YOLO models have set the benchmark for state-of-the-art methods of visual object detection. The latest version of YOLO object detection models—YOLOv7—was proposed in 2022 [79] and outperformed most well-known object detectors such as Cascade-Mask R-CNN [80], R-CNN [81], YOLOv4 [76], YOLOR [78], YOLOv5, YOLOX [82], PPYOLO [83], and DETR [84]. YOLOv7 reduces about 40% of the number of parameters and 50% of the computational costs of real-time object detection. YOLOv7 proposes the Extended Efficient Layer Aggregation Networks (E-ELAN) architecture to improve the self-learning ability of the network without destroying the original gradient path. It adopts a cascade-based model scaling method to be able to generate models of corresponding scales for practical tasks to meet detection requirements. YOLOv7 architecture is developed into six different tiers based on the complexity of the model and the number of parameters, with the basic model being YOLOv7, while the largest model is called YOLOv7-E6E. YOLOv7-E6E is an Extended-ELAN design incorporated into YOLOv7-E6. YOLOv7-E6 uses a newly proposed compound scale technique (scaling up the depth of the computational block by 1.5 times and the width of transition block by 1.25 times) on base models [79]. We trained both YOLOv7 and YOLOv7-E6E to compare the results and make further decisions on their suitability for the task to be performed.

### 2.4. Model Architecture and Training Process

The performance of a large DL model depends on various factors, including dataset size. Due to the higher contrast between the model size and model parameters, the first objective of this study was to identify the right model among the versions of YOLOv7 suitable for our study. To achieve this, we trained both YOLOv7 (basic) and YOLOv7-E6E (largest) using transfer learning techniques on our primary dataset. The model picked during the outcome of this study was then used for pseudo-labeling [85] and further analysis (Figure 3).

Training with images that include the background can provide the model with more contextual information and potentially improve its performance or provide unnecessary background noise and potentially degrade its performance on real-world images that contain cluttered or complex backgrounds. To compare and understand the influence of complex backgrounds such as a soybean field, where the background color can be monotonous and highly similar to the pod color, we trained YOLOv7 with and without background pixels for every image. With the assumption that training with images that have the background removed can simplify the training process and potentially lead to faster convergence, the goal of this training was also to compare the learning curve of the models under the removed and non-removed background datasets. Training YOLOv7 with images that have the background removed leaves only the objects of interest present in each image.

The YOLOv7 model was trained on background and no-background datasets using the transfer learning technique on an NVIDIA RTX A5000 Graphics Card for 300 epochs using the default hyper-parameters. A batch size of 16 with the Adam optimizer [86] was used in all of our training processes. To evaluate all three models, a test dataset with 150 images, which are new and diverse from the training and validation dataset is the comma necessary was created. This dataset was generated using pseudo-labeling [85] which a human annotator went through carefully to correct the mistakes and add missing annotations. These data contained previously unseen images at early stages of soybean growth, where in some cases the leaves are still present and the pod are still green in color. The model was evaluated based on mean Average Precision (mAP) at 50% and mAP at 50:95% at 65% Intersection over Union (IoU) threshold and the final model with the highest score was picked as the best output for this study. The steps taken in this study including data pre-processing, training process, and model evaluation are summarized in Figure 3.

## 3. Results

### 3.1. Main Model Evaluation

During model evaluation, it is important to report both training and test results since the model’s performance on the training set may not reflect its performance on new, unseen data in the real world. Thus, the results of both training and testing the model were evaluated and reported in this section, to find the best variant of YOLOv7 suitable for our study. Given 35,082 images used for training, the YOLOv7 model achieved a true positive rate of 97% while YOLOv7-E6E achieved 98%. The false negative rate for both the models was between 2% and 3%, meaning that the model failed to localize 2–3% of the objects it was trained to recognize. In terms of mean average precision (mAP), the models achieved a mAP@50 of 97.87% for YOLOv7, 98.7% for YOLOv7-E6E, a mAP@50:95 of 75.27% for YOLOv7, and 79.7% for YOLOv7-E6E.

Our primary training results indicate that both models demonstrated strong performance on the training datasets by accurately and precisely identifying objects. Notably, YOLOv7-E6E displayed slightly superior performance compared to the other model. However, it is important to note that the generalization ability of the model, or how well it performs on new data, cannot be determined from these results alone. It would be necessary to evaluate the model on a separate test set to determine its generalization ability. After evaluating the model on our test set (Table 2), the YOLOv7 model achieved a mAP@50 of 93.4% and a mAP@50:95 of 83.9% while attaining a precision of 89.4% and a recall of 94.7%. Similarly, the test output of YOLOv7-E6E exhibited similar but lower scores on both mAP50 and mAP50-95. Based on the results observed (Table 2), YOLOv7 performed better with an image input size of 640 × 640 while YOLOv7-E6E performed better when trained and tested with an image size of 1280 × 1280. Overall, these results suggest that the model was able to accurately identify and classify a large number of pods in the test set with a high level of precision and recall, making it a potential solution for pod-count-based soybean yield estimation. Therefore, YOLOv7 (the basic model) was chosen for the remaining analysis in this study.

The YOLOv7-E6E model, with its larger architecture size, is better suited for training on benchmark datasets with an image input size of 1280. This larger architecture allows YOLOv7-E6E to effectively process and extract features from images with more details and higher resolutions. In contrast, models like YOLOv7 are specifically designed to excel at detecting and learning from smaller image sizes, such as 640. These models are optimized to perform well when dealing with images that have moderate quality and relatively lower levels of detail. Our primary dataset consisted of 35,082 soybean images captured at a 1280 × 720 resolution. To simplify the images, the background was removed, resulting in scaled-down image details. It appears that YOLOv7-E6E may have been overfitting to our dataset, likely due to its larger size. Scaling the images up to 1280 introduced distortion and stretching, making pod detection more complex. However, when the images were scaled down to 640 and trained using a smaller architecture, all essential information was retained, enabling the smaller model to learn quickly. Our study suggests that the combination of YOLOv7 architecture, an image size of 640, and our dataset is optimal for in-field soybean pod detection and counting with an Intel^@^ RealSense^TM^ Depth Camera D405 (©Intel Corporation, Santa Clara, CA, USA).

Moving forward, our future work will focus on incorporating additional datasets from upcoming years to further validate this evaluation and improve our results.

### 3.2. Background vs. No-Background Model Evaluation

To better understand the impact of quantitative data on qualitative aspects of the model and to understand the effect of background on training, we compared the performance of YOLOv7 on two sub-datasets, one of which contained a background and the other did not. We performed transfer learning on two separate YOLOv7 weights, each trained on one of the sub-dataset. Figure 4 shows the learning curve of YOLOv7 during training with and without background.

In both algorithms, the output clearly demonstrated a much faster learning curve for models trained without the background, as indicated by higher mean average precision in both 50% and 50–90% thresholds. The graphs suggest that by removing the background, the DL, when trained with the same amount of data, manages to learn to localize the soybean pods much more quickly than when background noise is present. The outputs of testing the YOLOv7 model (trained on non-background images) showed a 10.2% improvement in precision, a 16.4% increase in recall, a 13.8% improvement in mAP@50, and finally a 17.7% improvement in mAP@.5:.95 thresholds (Table 3).

According to Table 3, the comparison between the model trained on the primary dataset (35,082 images) and the no-background dataset (1000 images) shows a clear difference in the model performance while testing on the new, unseen dataset (Figure 5). Based on this, it is concluded that a larger training dataset helps improve both the model’s accuracy and its overall generalization ability to a great extent. The YOLOv7 model trained on the primary dataset shows a 5% improvement in precision, an 11.1% increase in recall, a 4.2% improvement in mAP@50 and a 15.4% increase in mAP@50:95 over the model trained on the 1000 image dataset. While the time required to train YOLOv7 increases drastically with the increase in training data, the improvement in detection performance makes it well worth the extra time spent training the model.

After evaluating YOLOv7 (trained on the primary dataset) on the test dataset with continuous incremental IoU thresholds by 1% starting at 20% up to 95%, it was found that at an IoU threshold of 45–69%, the model resulted in the maximum mAP@0.5 score of 93.4% (Figure 6). Similarly, the highest mAP@0.5:95 scores of 84.0% were attained at IoU thresholds ranging from 45% to 58%; the highest precision and recall were observed at 24% and 81% IoU thresholds, respectively. The above output shows that the ideal IoU threshold for deploying the model would be at 58%, which is likely to output a more reliable pod count.

When a confidence threshold is introduced, a threshold value above which predictions are considered positive is set. Any prediction with a confidence score below the threshold is considered negative. By increasing the confidence threshold, we become more stringent in our acceptance of positive predictions.

As we increase the confidence threshold, the model becomes more cautious and conservative in labeling instances as positive. It only labels instances as positive when it is more confident about the correctness of its prediction. This cautiousness leads to a decrease in the number of false positives because the model is less likely to make positive predictions with lower confidence, reducing the chances of incorrect predictions.

With fewer false positives and a more conservative approach, the proportion of true positives among the positive predictions (i.e., precision) increases. Therefore, precision generally increases as we raise the confidence threshold because the model becomes more selective and confident in its positive predictions.

However, it is important to note that increasing the confidence threshold may also result in a decrease in recall (the proportion of true positives predicted correctly out of all actual positive instances). There is typically a trade-off between precision and recall, and adjusting the confidence threshold allows us to find a balance based on the specific requirements of the problem at hand.

Thus, we performed the same analysis, but instead the model was run at a constant 45% IoU threshold while varying the confidence threshold in the range of 10% to 90% at a step size of 1% (Figure 7).

To determine an optimized confidence threshold in practical applications, different techniques such as precision–recall curves (Precision–Recall: https://scikit-learn.org/stable/auto_examples/model_selection/plot_precision_recall.html, accessed on 1 February 2023) or the F1 score (F1 Score: https://scikit-learn.org/stable/modules/generated/sklearn.metrics.f1_score.html, accessed on 1 February 2023). Plotting a precision–recall curve can help visualize the trade-off between precision and recall at different confidence thresholds. The curve represents precision as a function of recall for various threshold values. Also, The F1 score is the harmonic mean of precision and recall, combining both metrics into a single value. It provides a measure of the overall performance of the model at a given threshold. In practical applications, the choice of an optimized confidence threshold depends on the specific requirements and constraints of the problem. If the cost of false positives is high, precision might be prioritized and a higher threshold should be selected. Conversely, if missing positive instances is more detrimental, recall be prioritized and a lower threshold should be selected.

Our data suggest that the trained model is able to hold an acceptable mAP@50 score of 91.3% and an mAP@50:95 of 83% up to a 34% confidence threshold. To reduce the number of false positive detections, we recommend the use of a confidence score of up to 34% during the model deployment for pod count in soybeans.

## 4. Discussion

To convert pod counts into a yield measurement, there are several factors that should be considered. Although studies have shown that there is a relationship between pod numbers and soybean yield, it is not a direct relationship [64,87,88,89]. Soybean yield is influenced by many factors, including plants per acre, pods per plant, seeds per pod, and seeds per pound (seed size) [90].

A higher number of pods per plant is associated with a higher soybean yield. However, the number of seeds per pod and the weight of each seed also play a role in determining yield. For example, a plant with a high number of small seeds per pod may have a lower yield than a plant with a lower number of larger seeds per pod, even if the plant with the larger seeds has fewer pods. Therefore, it is important to consider the number of pods per plant as well as other factors when predicting soybean yield. It is worth noting that counting the exact number of pods using digital methods is impossible because of occlusion. The focus of this paper was on counting the pods with a relatively high degree of accuracy. The key idea behind our proposed framework was to develop a non-destructive platform that is able to estimate the number of pods per plot in the shortest possible time with acceptable accuracy. Our main hypothesis was that, with a depth camera that removes background automatically, there is no need to force extra computational pressure on the framework to remove background, and thus the model could learn to count pods per image more accurately with a minimum number of false-negative and/or false-positive detection.

The results from this study indicate that both YOLOv7 models performed well on the training and test data, with high accuracy and precision in identifying objects, although the basic YOLOv7 model outperformed the YOLOv7-E6E. This is an important consideration when using the model for pod counting in soybeans, as the accuracy and precision of the model directly impact the reliability and usefulness of the predictions made. A high true positive rate (97% in this case) means that the model is able to correctly identify and locate a high percentage of the objects it was trained to recognize. This is important for pod counting, as it means that the model is likely to accurately identify and count the pods in an image. Similarly, a low false negative rate (3% in this case) means that the model is unlikely to miss any pods that are present in the image, which is also important for accurately counting the pods. The mean average precision (mAP) metric measures the overall accuracy of the model in identifying objects. The model’s mAP@50 of 97.87% and mAP@50:95 of 75.27% on the training data indicate that it is able to accurately identify a high percentage of the objects it was trained to recognize. Similarly, the model’s mAP@50 of 93.4% and mAP@50:95 of 83.9% on the test data indicate that it is able to generalize well to new data, suggesting that it is likely to accurately identify and count pods in new images.

The precision and recall results on the test set (89.4% precision and 94.7% recall) also indicate that the model is able to accurately identify and count the pods in the images. Precision measures the percentage of correctly identified objects out of all the objects identified by the model, while recall measures the percentage of correctly identified objects out of all the objects that are present in the image. High precision and recall values indicate that the model is able to accurately identify and count the pods in the images. The above results suggest that the YOLOv7 model is well-suited for predicting the number of soybean pods per plot accurately and reliably; therefore, we suggest that the YOLOv7 model has strong potential for use in this application.

When testing the YOLOv7-E6E model, the mAP@50 and mAP@0.5:.95 decreased by 5.5% and 22.7%, respectively. This is a significant decrease in model performance despite a larger model size and a significantly higher number of parameters. In the best case, when the model is tested with an image size of 1280, YOLOv7-E6E still shows a 1.4% lower performance (mAP@50) compared to the basic model. While the model exhibits a higher mAP@50 during training (98.7%), its poor performance on new and unseen testing data shows an overfitting problem. The occurrence of overfitting in our study could be attributed to the extensive architecture of YOLOv7-E6E. This architecture allows the model to becomes overly specialized for the problem at hand, particularly considering that our dataset comprises only 35,082 images. In addition to the less favorable performance of YOLOv7-E6E on our dataset, challenges arise even if the model is to be deployed in the field for real-time applications due to its larger model size. Further comparative analysis with an increased dataset needs to be carried out before ruling out the use of YOLOv7-E6E for soybean pod count.

Our current analysis only implements a DL algorithm evaluation based on the COCO evaluation metrics used to evaluate the algorithm itself. The main objective of this study was to evaluate the impact of removing the background from a DL-based object detection algorithm such as YOLOv7 by using a depth camera. In addition to this, we also compared YOLOv7 and YOLOv7-E6E and selected the best model fit for counting soybean pods for breeding assistance. Our future work will focus on testing and improving this system on-field and applying it to soybean pod count for yield prediction. While multi-view imaging does not contribute significantly toward yield prediction, single-view images from both sides of the plots are necessary [52]. Therefore, our platform will be expanded to equip cameras on both sides to capture pods that are otherwise missed due to occlusion. Also, our current method of data capture is set to a fixed interval at 4 Hz, which means the speed at which the platform is moved affects the number of images captured per plot. Moreover, the current method does not provide a means to count soybean pods per row but rather helps estimate the average pod count density per plot. Future work will hence be focused on using an RTK GPS to trigger the image capture with a predefined overlap or using panoramic images to capture each plot as a single image to address the mentioned problems.

In the next phase of this study, in order to develop a framework that is able to estimate the number of soybean pods using non-destructive proximal methods, there are several points that should be considered, including:Our current platform only uses one depth camera for image acquisition from plots and captures only one side of the crop row, which means that the pods on the other side are not captured. A potential solution could be using two depth cameras on both sides of the crop rows [52]; however, then, the issue of counting the same pods twice arises.Our approach counts the average number of pods per image in a row, which is not accurate when it is desired to estimate the yield using the number of pods per plot. For a much more accurate pod count estimation, all the factors associated with yield, such as, the number of plants per row or area, the number of pods per plant [13], the number of seeds per pod [23], and the seed size [91] need to be measured and calculated.Two-stage object detection frameworks need to be researched to find the difference in speed and performance during the pod count process.Despite the efforts to remove background noise from the images through the depth-based segmentation method, the lower 20% of the image is saturated with background noises such as fallen leaves, soil, etc. This needs to be resolved by optimizing the camera position or developing another background segmentation method.

## 5. Conclusions

This study aimed to improve the accuracy of soybean pod counting by using a depth camera to filter out the background of RGB images in real time to overcome the background issue and improve the performance of the classification model. Object detection models YOLOv7 and YOLOv7-E6E were compared, and YOLOv7 was found to be the best model suitable for counting soybean pods. We then carried out a comparative analysis of the model’s performance by training the ML model before and after removing the background from the images. Removing the background using a depth camera significantly improved the overall performance of YOLOv7, resulting in an increase in performance compared to when the background was present. The results of this study indicate that using a depth camera to remove the background and the YOLOv7 algorithm for pod detection can significantly improve the performance of soybean pod counting and this method has the potential to be a useful tool for soybean breeders to estimate soybean yield from a large number of genotypes prior to harvest.

## Figures and Tables

**Figure 1 sensors-23-06506-f001:**
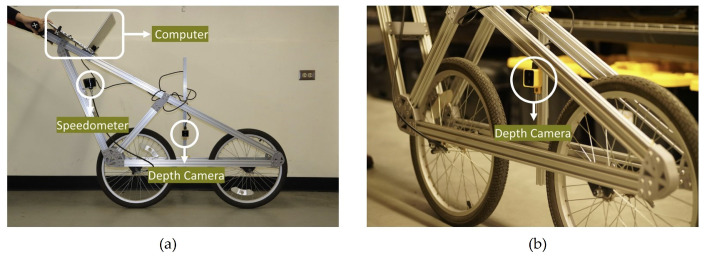
(**a**) Details of the bicycle-like platform used in this study. (**b**) A view of the camera installed on the platform for image collection. (**c**) A schematic representation of the field data collection.

**Figure 2 sensors-23-06506-f002:**
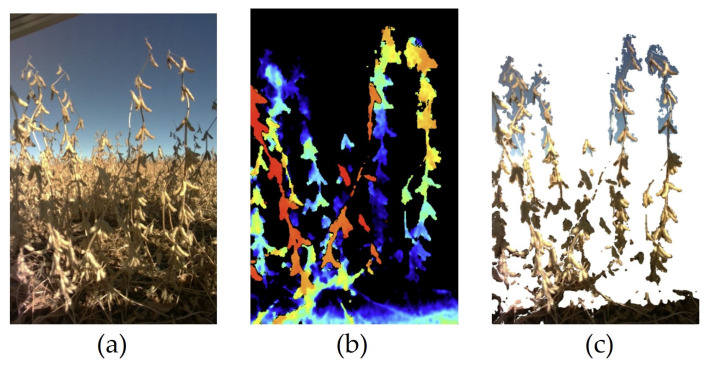
Sample images from the field test dataset. These three images correspond to the same view within a plot: (**a**) RGB image collected with depth camera in soybean field. (**b**) The depth image defines the color intensity for the seen objects with respect to their distance from the camera, with red representing the closest objects and blue the farthest. (**c**) All objects captured in the RGB image that are farther than 50 cm are removed.

**Figure 3 sensors-23-06506-f003:**
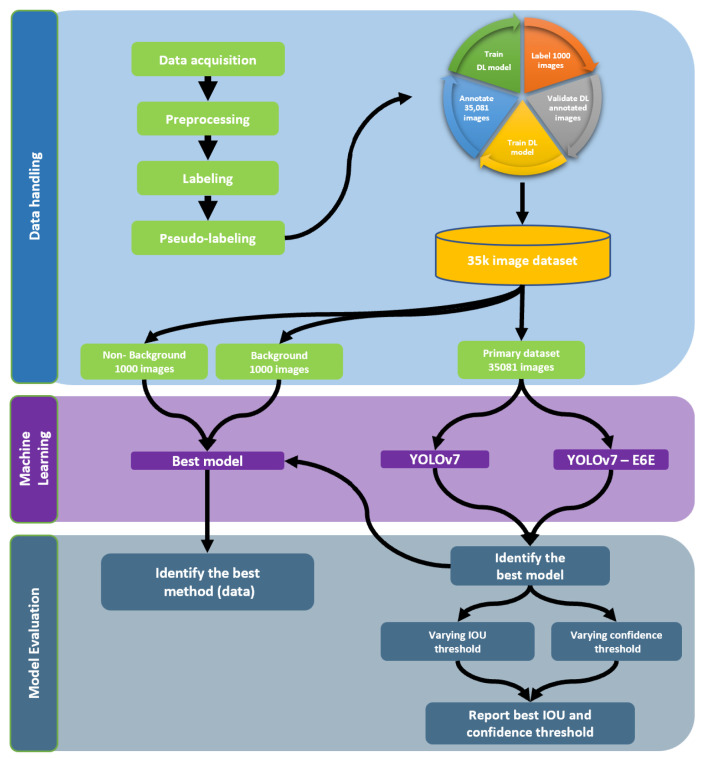
Flowchart of the steps taken from data acquisition to model evaluation.

**Figure 4 sensors-23-06506-f004:**
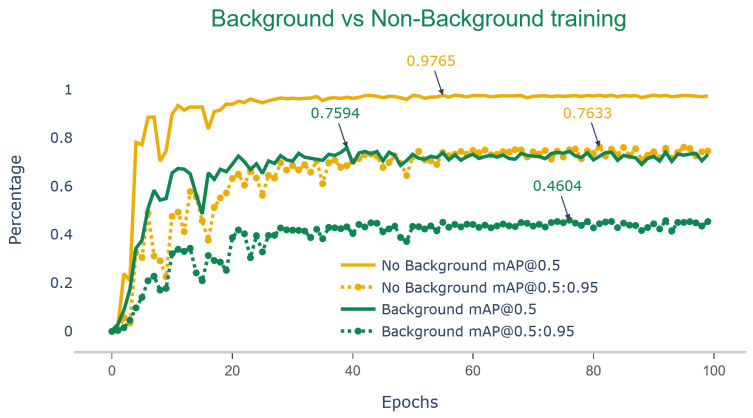
Metrics of mAP@50 and mAP@50:95 comparison for soybean datasets with images containing background vs. background segmented during training of YOLOv7 for 300 epochs (trained with the same parameters).

**Figure 5 sensors-23-06506-f005:**
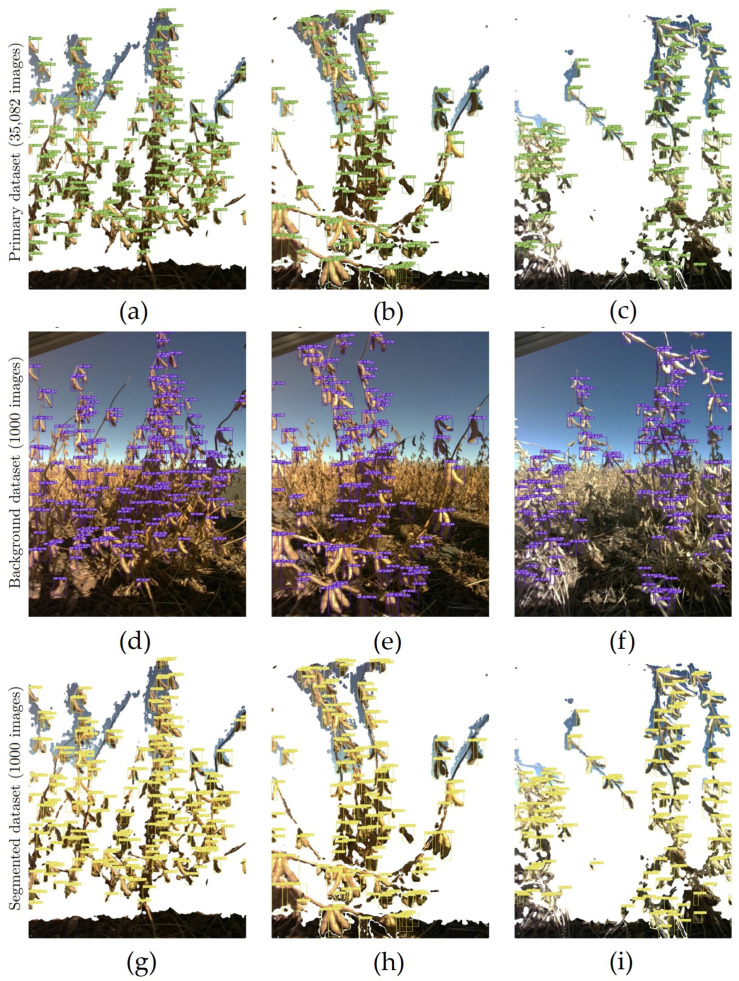
Comparison of soybean pod detection accuracy using YOLOv7 trained on primary dataset (first row), images with background (second row), and segmented background images (last row). The ground truth pod count and the predictions by the model for each image is given: (**a**) Ground Truth: 203 pods, DL Model Prediction: 208 pods; (**b**) Ground Truth: 135 pods, DL Model Prediction: 132 pods; (**c**) Ground Truth: 120 pods, DL Model Prediction: 136 pods; (**d**) Ground Truth: 203 pods, DL Model Prediction: 212 pods; (**e**) Ground Truth: 135 pods, DL Model Prediction: 157 pods; (**f**) Ground Truth: 120 pods, DL Model Prediction: 176 pods; (**g**) Ground Truth: 203 pods, DL Model Prediction: 223 pods; (**h**) Ground Truth: 135 pods, DL Model Prediction: 146 pods; (**i**) Ground Truth: 120 pods, DL Model Prediction: 154 pods.

**Figure 6 sensors-23-06506-f006:**
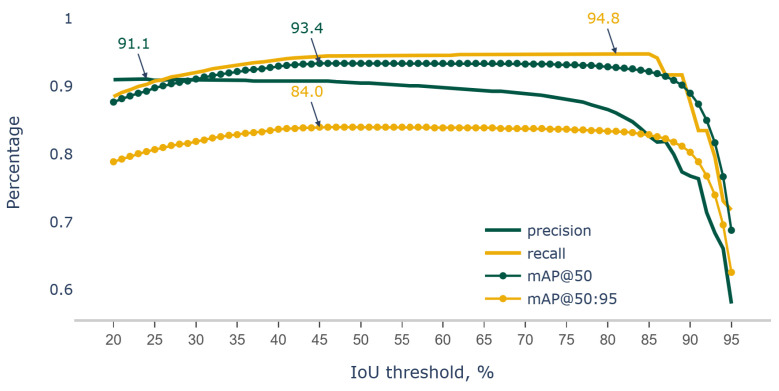
Metrics of precision, recall, mAP@50 and mAP@50:95 comparison for YOLOv7 trained on soybean primary datasets evaluated on test dataset at IoU threshold varying for 20 to 95 percentages.

**Figure 7 sensors-23-06506-f007:**
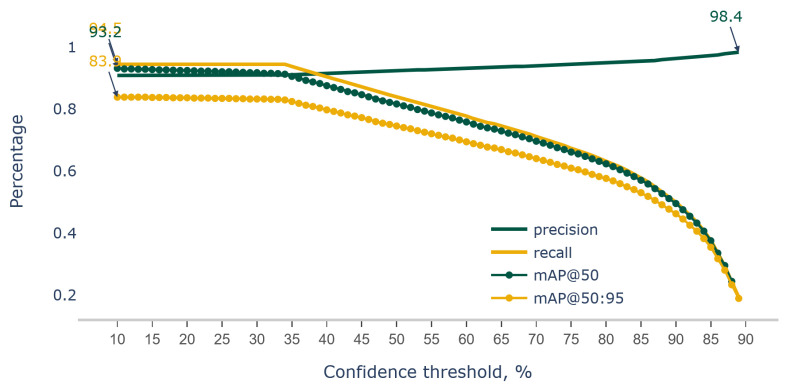
Metrics of precision, recall, mAP@50 and mAP@50:95 comparison for YOLOv7 trained on Soybean primary datasets evaluated on test dataset at confidence threshold varying for 10 to 90 percentages.

**Table 1 sensors-23-06506-t001:** Basic information of the soybean trail used in this study.

Trial ID	Number of Plots	Plot Size (Width × Height)	Plot Spacing
Conv–2–row	190	3.0 m × 18.2 m	0.76 m
PR–2–row	167	6.0 m × 57.9 m	0.76 m
Conv–4–row	357	3.0 m × 16.7 m	0.76 m
PR–2–row	45	3.0 m × 9.1 m	0.76 m

**Table 2 sensors-23-06506-t002:** Comparing the testing output of YOLOv7 and YOLOv7-E6E under different parameters under 1% confidence threshold and 65% IoU threshold.

Model	Image Size	Precision	Recall	mAP@.5	mAP@.5:.95
YOLOv7	640	89.4%	94.7%	93.4%	83.9%
	1280	68.8%	72.4%	74.9%	51.3%
YOLOv7-E6E	640	83.9%	81.9%	87.9%	61.2%
	1280	87.6%	86.5%	92.0%	73.3%

**Table 3 sensors-23-06506-t003:** Performance metrics of testing output of the YOLOv7 model trained on three soybean image datasets.

Dataset	No. Images	No. Training Images	Precision	Recall	mAP@.5	mAP@.5:.95
Primary	35,082	31,578	89.4%	94.7%	93.4%	83.9%
No Background	1000	900	84.4%	83.6%	89.2%	68.5%
Background	1000	900	74.2%	67.2%	75.4%	47.8%

## Data Availability

The datasets generated and analyzed for this study can be found in the Github repository Soybean pod count depth segmentation project 2022 accessible at https://github.com/jithin8mathew/soybean_pod_count_Depth_segmentation_project (accessed on 28 June 2023).

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
