# Peer review of "A Novel Approach to Pod Count Estimation Using a Depth Camera in Support of Soybean Breeding Applications"

_sensors, 2023, doi:10.3390/s23146506_

Round 1

Reviewer 1 Report

This study aimed to improve the accuracy of soybean pod-counting by using a depth camera to filter out the background of RGB images in real-time to overcome the background 441 issue and improve the performance of the classification model.

 the Quality of English Language is good.

Author Response

Dear reviewer,

Thank you so much for spending your invaluable time to review our manuscript and for the comments and feedback you have provided us with. Please find the attached file.

Regards,

Authors

Reviewer 2 Report

Pod-count is an important task for the yield estimation of soybean. This manuscript presented an automatic pod counting by the depth camera and deep learning model. It is a novel application in pod counting in field conditions. A depth camera is utilized to remove the background image and the false detection could be significantly reduced. 

This manuscript could be improved by considering the following issues:

1. The contributions of this work should be emphasized. Instead of the application of depth camera, I suggest that the automatic counting of pod in field conditions might be more interesting to the readers. Automatic detection combining with the bicycle-like platform shows novelty in soybean field monitoring.

2. An experiment section should be given to describe the experimental conditions and process, including hardware, software setups and parameters.

3. Data description in the manuscript is insufficient and unclear. A data description table should be given that presents the number of images in each datasets, and the number of pods. And also the number of images in the training set and testing set. 

4. Background removal of plant images has been presented in the previous work, please add the following work.

Xia, C., Wang, L., Chung, B. K., & Lee, J. M. (2015). In situ 3D segmentation of individual plant leaves using a RGB-D camera for agricultural automation. Sensors, 15(8), 20463-20479.

5. Only YOLOv7 is involved in the tests. The authors might consider compare with other SOTA methods.

6. In table 2, detection performance was significantly decreased in YOLOv7 with 1280 image size. why

7. In Fig.7, you should briefly explain why precision increases with the confidence threshold. And, how to determine an optimized confidence threshold in practical applications?

8. MAPA seems to be a typo.

9. Public your data could increase the impact of your work. It is inaccessible yet.

Author Response

Dear reviewer,

Thank you so much for spending your invaluable time to review our manuscript and for the comments and feedback you have provided us with. Kindly find the attached file in reponse to your comments. 

Regards,

Authors

Reviewer 3 Report

-Intro is strong but explicit contribution to literature needs listing

-Model architecture, algorithm and flowchart could help present work better perhaps?

-I note that the accuracy metric is not used? How come?

-The conclusion should carry notes of future work

Well written paper

n/a

Author Response

(The authors gave the same response as above.)

Round 2

Reviewer 2 Report

I have confirmed the contents of the manuscript and I'd like to suggest accepting the manuscript for publication.